# Transforming students' attitudes towards learning through the use of successful educational actions

**Javier Díez-Palomar**[1]*, **Rocío García-Carrión**[2], **Linda Hargreaves**[3], **María Vieites**[4]

**1** Department of Linguistic and Literary Education, and Teaching and Learning of Experimental Sciences and Mathematics, University of Barcelona, Barcelona, Spain, **2** Faculty of Psychology and Education, University of Deusto, Ikerbasque, Basque Foundation for Science, Bilbao, Spain, **3** Faculty of Education, University of Cambridge, Cambridge, United Kingdom, **4** CREA–Community of Research of Excellence for All, University of Barcelona, Barcelona, Spain

* jdiezpalomar@ub.edu

**Data Availability Statement:** All relevant data are within the manuscript and its Supporting Information files.

## Abstract

Previous research shows that there is a correlation between attitudes and academic achievement. In this article, we analyze for the first time the impact of interactive groups (IG) and dialogic literary gatherings (DLG) on the attitudes that students show towards learning. A quantitative approach has been performed using attitude tests validated by previous research. The data suggest that in both cases, the participants show positive attitudes. The social context has an important influence on students' attitudes. The items with higher correlations include group work, mutual support, and distributed cognition. In the case of IGs, group work is much more appreciated, while in the case of DLGs, self-image and self-confidence are the two most clearly valued attitudes. The positive impact of IGs and DLGs on students' attitudes may have potential for teachers in transforming their practices and decision-making within the classroom.

## Introduction

In this article, we address the following research question: What impact does participation in interactive groups (IGs) and dialogic literary gatherings (DLGs) have on the attitudes that students show towards learning? To define "attitudes", we draw on the definition of Harmon-Jones, Harmon-Jones, Amodio and Gable [1], who characterize attitudes as "subjective evaluations that range from good to bad that are represented in memory" [1]. This definition is also consistent with the classic definitions about "attitudes" used in social psychology studies [2].

Previous research suggests that there is a clear relationship between students' attitudes and their academic achievement [3,4]. Decades ago, classic studies in the field of educational research [5] concluded that teachers' expectations about students' attitudes and behaviors may explain students' effective academic achievement. According to [4], the process of the "social construction of identity" explains why there are students who seem destined to obtain poor academic results. Drawing on the theoretical approach developed by Mead [6], Molina and her

**Funding:** JDP, RGC, LH and MVC want to acknowledge the funding provided by the EU Commission under the grant num. 2015-1-ES01-KA201-016327, corresponding to the project Schools as Learning Communities in Europe: Successful Educational Actions for all, (SEAS4ALL), under the program ERASMUS +; and the Spanish Ramón y Cajal Grant RYC-2016-20967 for open access publication of the article.

**Competing interests:** No authors have competing interests.

colleagues [4] argue that the way in which a student defines his/her own identity determines his/her own learning expectations and, as a consequence, his/her own academic career. In this sense, the process of constructing identity is social in essence.

According to Mead [6], the self emerges as a result of social interaction with other people who project their expectations and attitudes on the individual. The identity of a person is formed by two components. The first component is the *me*, which is of social origin and incorporates the attitudes of others about the individual; the second component is the *I*, which is the conscious reaction of each individual to those attitudes. A process is thus created in which identity is the result of the dialogue between the individual and others. This somehow explains why some students end up developing an identity as bad students, while others develop an identity as good students. This process has been called the "Pygmalion effect" by educational researchers [5]. As Flecha [7] suggests, drawing on successful educational actions (SEAs), teachers get their students to achieve better results, and that, in turn, explains why these students improve their self-concept as students (i.e., their *me* and *I*, in Mead's terms). However, does that mean that they also change their attitudes towards learning in school?

Classic studies such as *Learning to Work* [8] suggest that students with low academic performance tend to be children who reject school. These students tend to manifest that feeling of rejection in wayward attitudes. These children also do not see school as a desirable or attractive place. In contrast, they show an attitude of rejection and resistance towards schooling, which is accompanied by low academic performance. Later researchers, such as Bruner, have suggested that this attitude is the result of the failure of the schools to respond to the expectations of these children (and their families) [9]. Students' identities are defined in other spaces and with other references. This may have a negative impact on the ability of teachers to teach. Various studies have suggested that as some of these children grow, their interest in school diminishes. This happens especially in the transition between primary and secondary school, as some students lose interest in science, mathematics and other subjects. Given this situation, researchers have found that learning initiatives located outside of the school can change these attitudes towards learning, as in the case of visiting museums, laboratories, or research centers [10].

This article discusses the impact of participating in two educational activities previously defined as successful educational actions (SEAs) [7] on attitudes towards learning shown by students who have participated in these actions. Thus far, we have clear evidence of the positive impact that SEAs have on learning outcomes [11–15], and there are studies suggesting that there is also a positive effect on the coexistence and cohesion of the group-class [16,17]. However, previous studies have not explored the impact that such SEAs may have on students' own attitudes. Therefore, this article discusses this dimension of learning, which, as the studies mentioned above claim, is a relevant aspect to understanding how learning works.

## Theoretical framework

### Attitudes and learning

There is an assumed belief in education that there is a direct relationship between student attitudes and academic achievement. Renaud [18] distinguishes between dispositions and attitudes by stating that the former is more "resistant" to change than the latter. Dispositions are defined as "more general and enduring characteristics", while attitudes are tendencies or internal states of the person towards anything that a person can evaluate, such as "learning math, extracurricular activities, or the general notion of going to school" [18]. Previous research that exists on attitudes and learning has found that there is a clear relationship between both aspects. Renaud [18] quotes literature reviews that indicate that there is a correlation between

attitude and achievement in mathematics [19–21] and in science [22]. According to previous research, the relationship becomes stronger at higher educational levels [22–24].

Ma and Kishor [19] analyze three indicators that refer to attitude to evaluate their impact on academic achievement: the self-concept about mathematics, the support of the family and the gender role in mathematics. According to their data, the most important correlation corresponds to the self-concept (p. 24). Masgoret and Gardner [23] found that motivation is more closely related to academic achievement than attitude. Attitude, on the other hand, seems to be related to achievement; however, it is related indirectly through motivation. Motivation and self-concept present a clear relationship (a correlation exists), but the research is not conclusive in regard to which direction the relationship works, i.e., we do not know (yet) if it is the motivation that gives rise to the person's positive self-concept or if having a positive self-concept translates into an increase in terms of motivation. In any case, both variables seem to correlate directly with academic achievement; higher motivation and self-concept are associated with better academic results (in general terms).

## The "symbolic interactionism" approach

The theoretical approach that has devoted the most effort to analyzing the relationship between attitudes, motivation, self-concept and academic achievement is that of symbolic interactionism. George H. Mead [6] is one of the best-known representatives of this theory. According to his findings, self-concept is of social origin. "The self is something which has a development; it is not initially there at birth" [6]. To explain this process, Mead proposes the concept of the generalized other. According to him, the generalized other is "the organized community or social group which gives to the individual his or her unity of self" [6]. Mead illustrates how this concept works to create the self-concept by drawing on the game metaphor. He uses the example of baseball. The baseball team is what Mead calls the generalized other. Each player has a specific role within the team, and he or she acts in accordance with what is expected of him or her in that position. The rest of the team does the same, so that individual actions are defined and carried out within the more global unit that defined is the team (i.e., the generalized other). Using this example and others, Mead [6] was able to show that the self is the result of a social process. Similarly, Vygotsky [25] claimed that higher psychological functions emerge through interpersonal connections and actions with the social environment until they are internalized by the individual.

Mead states as follows:

It is in the form of the generalized other that the social process influences the behavior of the individuals involved in it and carrying it on, i.e., that the community exercises control over the conduct of its individual members; for it is in this form that the social process or community enters as a determining factor into the individual's thinking. [6]

This social process involves interaction with other individuals in the group through shared activities. The classroom is the perfect example of a group. The teacher and the students are part of a social group with defined norms [26] as well as an institutional objective (teaching and learning), where each "player" performs a specific role according to those (declared or implicit) norms.

In some investigations, this social unit has been defined as a "community of practice" [27,28]. Brousseau [29] uses the concept of "contracte didactique" to characterize this social unit and analyze its functioning in the mathematics classroom. According to Brousseau, there

is a relationship between the different actors (individuals) participating in the mathematics classroom, in which each plays a specific role and has specific responsibilities. The teacher has the obligation to create sufficient conditions for the appropriation of knowledge by the students and must be able to recognize when this happens. Similarly, the responsibility of the students is to satisfy the conditions created by the teacher. Brousseau studies how the teacher performs what he calls the didactic transposition of scientific knowledge to be taught in school. S/he has to identify the epistemological obstacles and the cognitive obstacles that make it difficult or students to learn in the classroom.

However, other studies have suggested that there are factors of another nature (neither cognitive nor epistemological) that also influence the academic achievements reached by students. This is the case with interactions [16,30,31]. As Mead [6] suggested, the self-concept created by an individual is the result of the internalization of the expectations that each individual has of himself or herself by the role s/he plays in the group to which s/he belongs. For example, the student who always tries hard and answers the teacher's questions is fulfilling his or her role within the good student group. The group expects him or her to play that role. It is part of his/her identity. In addition, s/he acts accordingly. The effect of the positive or negative projection of expectations on students has been widely studied in education [32,33]. What we know is that teachers have to be cautious and try not to project negative expectations on students because that has a clear effect on their academic achievements, giving rise to well-studied interactions such as the Pygmalion effect, or the self-fulfilling prophecy [5].

However, the impact of successful educational actions [7] on the attitudes that students have towards learning at school in the context of interactive groups and dialogic literary gatherings has not been studied so far. This impact is what is discussed in this article.

## The successful educational actions of interactive groups and dialogic literary gatherings

The research question discussed in this article is framed in the context of the implementation of two successful educational actions identified by the European Commission in the research project titled *INCLUD-ED*: *Strategies for inclusion and social cohesion from education in Europe* [34]. A successful educational action is defined as an action carried out in the school, the result of which significantly improves students' learning [7]. The two successful educational actions that are discussed herein are interactive groups and dialogic literary gatherings.

**Interactive groups.** Interactive groups (IGs) consist of a particular group-based teaching practice in which students are put together in small groups of approximately six or seven students, with an adult person facilitating the task. IGs must be heterogeneous in terms of their composition, including children with different ability levels, gender, socioeconomic background, etc. The adult person (the facilitator) is a volunteer who encourages dialogic interaction among the group members while performing the task designed by the teacher. The teacher splits the students among four or five IGs (depending on the number of children in the classroom and the time available for the lesson). Each group of students has a task assigned, which the teachers have previously designed. There is a total of four or five tasks (the same number as the number of IGs). The assignments are about the subject that is being focused on in the lesson plan (i.e., mathematics, language, science, history, etc.). To perform the assigned task, groups have fifteen or twenty minutes of time (depending on the total time allocated for that activity in the school day). After this interval, the teacher asks the students to move to the next IG, where they will find another task. When the class is over, each of the children must have gone through all of the tasks. All of the children perform four or five different tasks

designed by the teacher to cover the curriculum requirements. In some schools, the kids move from one IG to the next. In other cases, the teachers prefer to ask the facilitators to move between the groups to avoid the noise and disorder created by the children getting up and moving to the next table (task).

The facilitators never provide solutions to the tasks executed by the students. Instead, they encourage students to share, justify, explain, their work to their group mates. Their responsibility is encouraging students to use dialogic talk [15], which is based on the principles of dialogic learning [35]. Research evidence on IGs suggests that using dialogic talk increases participants' chances of improving their academic achievements [15,30,36]. When students are asked by the facilitator to justify their answers to a task, they need to conceptually defend their claims; this implies that they must be able to not only understand the concept or concepts embedded in the assigned tasks but also explain them to their group mates. The type of talk (speech) that appears when children engage in this type of interaction has been defined as dialogic talk [30] because it is oriented towards validity claims [37], not towards the power position that children occupy within the group.

**Dialogic literary gatherings.** Dialogic literary gatherings (DLGs) are spaces in which students sit in a circle and share the reading of a classic literary book. The gathering is facilitated by the teacher, whose role is not to intervene or give his/her opinion, but to organize the students' participation by assigning them turns. Every child who wants to share his/her reading raises his/her hand and waits until the teacher gives him/her a turn. Readings come from classic literature, such as works by Shakespeare, Cervantes, Kafka, Tagore, etc. [38,39]. Students read at home the assigned number of pages (it either could be a whole chapter or a certain number of pages, according to the teacher's criteria). When reading the assignment, the student highlights a paragraph and writes down the reason for his/her choice. Then, during the DLG session in school, the children bring the paragraph or paragraphs they want to share with the rest of their classmates. At the beginning of the session, the teacher asks who wants to share his/her paragraph. S/he writes down the name of the students offering to share on a list. Then, the teacher starts with the first name on the list and that student reads aloud his/her paragraph; s/he also identifies on which page of the text it is so that the rest of the participants in the DLG can follow the reading and explains the reason for his/her choice. After the reading, the teacher opens the floor for questions. S/he always prioritizes the children who participate less often. When the teacher considers that the topic has been sufficiently commented on either because the idea that led to the intervention has been fully commented on or because the children's questions drifted to other irrelevant topics, s/he moves to the next name on the list. The process is repeated until the session ends.

Children, when talking about their paragraph, become involved in a process called "dialogical reading" [40], which is based on the application of Bakhtin's concept of "dialogism" [41]. Bakhtin explains this concept using the idea of "polyphony" to refer to the use of multiple voices in a narrative work, such as the case of Dostoevsky's poetics [41]. According to Bakhtin, no voice is the result of a single speech but rather it integrates different voices. This concept has been reused and reinterpreted in educational research. Drawing on those authors, knowledge is the result of internalizing the voices of multiple people (teachers, family members, friends, classmates, and other people) that we have encountered throughout our lives. DLGs recreate that multiplicity of voices through the dialogues that generate a space in which all children contribute with their opinions, ideas, and understandings about the paragraph being discussed. In this sense, reading understanding develops in a much deeper way than if the child had to read the material individually because s/he can incorporate the points of view of his/her peers into his/her own final comprehension.

## Methodology

The data used to discuss the research question come from a research project titled *SEAs4All–Schools as Learning Communities in Europe*. The dataset has been submitted to this journal as supporting data for public use. Six schools from the four European countries of Cyprus, United Kingdom, Italy and Spain participated in this project. Five of them were primary schools, and the last one was a middle/high school. All of the schools were selected because they applied successful educational actions (SEAs) [7]. After implementing IGs and DLGs, a survey was conducted in three of the schools to evaluate the impact of using these two types SEAs on students' attitudes and perceptions towards learning. Children between 7 and 11 years old participated in the survey. Two of the surveyed schools are located in the United Kingdom, and the third one is located in Italy. All of the schools are located in different contexts. One of the schools in the United Kingdom is in an area where families have a high economic status and high cultural capital (Cambridge), while the other English school is located in a neighborhood considered to be of a medium-level SES (Norwich). The Italian school is located in a low SES area of Naples. A total of 418 children participated in the survey (251 participating in DLGs and 167 engaging in IGs), as shown in Table 1.

To collect the data, the SAM questionnaire, developed at the Universities of Leicester and Cambridge, UK, was used as a model for the evaluation of the impact of the implemented educational actions. The original SAM questionnaire consists of 17 items that are measured using a 5-point Likert scale, which ranges between "strongly agree" and "strongly disagree." The questionnaire used in the current study was amended by drawing on previous results from a pilot test and was thus reduced to 12 items [S1 File].

The children answered a paper version of the questionnaire. The data were then coded and entered into an Excel matrix that was later used to analyze the data in SPSS (version 25.0). To debug the database and detect possible errors in the transcription, univariate descriptive analysis was conducted using the table of frequencies for each item to check that all codes and weights were aligned with the data collected through the paper questionnaires. Whenever an anomaly was detected, we proceeded to review the original questionnaire on paper to verify the information and data transcribed in the matrix.

To analyze the data, a descriptive report was first made by tabulating the data in frequency tables using the mean, median and mode, as well as the variance and standard deviation.

To answer the research question, a main components analysis was used because we do not know a priori if there is any explanatory factor structure that relates the IGs and DLGs with students' attitudes. To test the validity of using such analysis, Bartlett's sphericity test ($\chi^2$) was used to check the homoscedasticity of the three schools participating in the survey as follows:

$$\mathcal{X}^2 = \frac{(N-k)\ln\left(S_p^2\right) - \sum_{i=1}^{k}(n_i - 1)\ln(S_i^2)}{1 + \frac{1}{3(k-1)}\left(\sum_{i=1}^{k}\left(\frac{1}{n_i-1}\right) - \frac{1}{N-k}\right)} \tag{1}$$

where $k = 3$ represents each of the samples of the three schools whose students participated in the survey. Bartlett's test was applied both to the case of the subsample of students participating

**Table 1. Samples collected in the three schools participating in the survey.**

| | |
|---|---|
| School 1 (Italy) | n = 168 |
| School 2 (UK) | n = 29 |
| School 3 (UK) | n = 221 |
| Total | n = 418 |

in the DLGs and to the case of those participating in the IGs. The results of this test are shown in Table 2.

Before performing Bartlett's test, four of the items were recoded (#1, #3, #4, and #6) since the grading vector of the Likert scale used in these three items went in the opposite direction as that used for the rest of the items. These four items were displayed in a negative tone (i.e., "we learn best when the teacher tells us what to do", "learning through discussion in class is confusing", "sometimes, learning in school is boring", and "I would rather think for myself than hear other people's ideas"), unlike the rest of the items in which the tone of the answers was positive. Therefore, the labels of "strongly agree", "agree a little", "not sure", "disagree a little", and "strongly disagree" for items #1, #3, #4, and #6 referenced to a scale with a negative associated vector, whereas for the rest of the items, the same labels refer to a positive vector. For this reason, the responses of these four variables were recoded into four new variables that reversed the original direction of the response vector.

In both cases, (IGs and DLGs), Bartlett's test suggests that we can accept the null hypothesis, which means that we can use factor analysis to discriminate which principal components are the ones that explain the greatest percentage of the variance. The results of this analysis are discussed below.

### Ethic statement

The studies involving human participants were reviewed and approved by Ethics Committee of the Community of Research on Excellence for All, University of Barcelona. Schools collected the families' informed consent approving the participation of their children in this study.

The Ethics Board was composed by: Dr. Marta Soler (president), who has expertise within the evaluation of projects from the European Framework Program of Research of the European Union and of European projects in the area of ethics; Dr. Teresa Sordé, who has expertise within the evaluation of projects from the European Framework Program of Research and is a researcher of Roma studies; Dr. Patricia Melgar, a founding member of the Catalan Platform against gender violence and researcher within the area of gender and gender violence; Dr. Sandra Racionero, a former secretary and member of the Ethics Board at Loyola University Andalusia (2016–2018); Dr. Cristina Pulido, an expert in data protection policies and child protection in research and communication and researcher of communication studies; Dr. Oriol Rios, a founding member of the "Men in Dialogue" association, a researcher within the area of masculinities, as well as an editor of "Masculinities and Social Change," a journal indexed in WoS and Scopus; and Dr. Esther Oliver, who has expertise within the evaluation of projects from the European Framework Program of Research and is a researcher within the area of gender violence.

## Results

### Students' attitudes towards learning

The data collected suggest that the students who participate in the IGs and the DLGs have positive attitudes towards learning in general terms. Table 3 indicates that the answers for almost

**Table 2. KMO and Bartlett's test of sphericity.**

|  |  | IGs | DLGs |
|---|---|---|---|
| **Kaiser-Meyer-Olkin Measure of Sampling Adequacy** |  | 0,517 | 0,610 |
| **Bartlett's Test of Sphericity** | Approx. Chi-square | 30,576 | 98,833 |
|  | df | 6 | 28 |
|  | Sig. | 0,000 | 0,000 |

**Table 3. Valid percent of students' answers participating in IGs and DLGs.**

| Item | Strongly agree | Agree a little | Not Sure | Disagree a little | Strongly disagree |
|---|---|---|---|---|---|
| #1. We learn best when the teacher tells us what to do | 62,1 | 20 | 9,9 | 3,9 | 4,2 |
| #2. We learn more when we can express our own ideas | 48,8 | 25,6 | 18,5 | 2,5 | 4,7 |
| #3. Learning through discussion in class is confusing | 19,7 | 20,7 | 15,5 | 17,2 | 26,8 |
| #4. Sometimes learning in school is boring | 23,1 | 18,7 | 7 | 10,9 | 40,3 |
| #5. Learning in school is better when we have other adults to work with us | 62,4 | 14,5 | 10,1 | 5,4 | 7,6 |
| #6. I would rather think for myself than hear other people's ideas | 18,9 | 10,8 | 21,4 | 11,3 | 37,6 |
| #7. I enjoy learning when my friends help me | 65 | 13,6 | 10 | 4,6 | 6,8 |
| #8. It is good to hear other people's ideas | 55,9 | 22,4 | 9 | 3,7 | 9 |
| #9. Helping my friends has helped me to understand things better | 58 | 18,7 | 11,9 | 5,3 | 6,1 |
| #10. I am more confident about learning in school than I used to be | 61,8 | 15,4 | 14 | 2,5 | 6,4 |
| #11. I like discussing the books we read with the class | 46,9 | 20,3 | 12,1 | 7,3 | 13,6 |
| #12. At home, sometimes we talk about what I have been learning in school | 60,4 | 17,1 | 9,3 | 4,9 | 8,3 |

all the items are clearly positive; this is true for between 75% and 80% of the responses, except for three items (#3, #4, and #6). This outcome is understandable since in these three items, the interviewer changed the meaning of the question, i.e., instead of phrasing the questions positively, as with the rest of the items, the questions were phrased negatively, with the expected outcome being that the positive trend in the answers would be reversed, which is what occurred. Surprisingly, in the case of item #1, which we expected to function similar to items #3, #4 and #6, the responses are aligned with the rest of the items.

The answers to item 1 ("We learn best when the teacher tells us what to do") may indicate an active role by the teacher, which a priori would not be the expected answer in the context of using IGs and DLGs. In contrast, what we would expect in that context is for students to show a preference for answers related to an active role of the student, which is the case for the rest of the items analyzed. However, the fact that the respondents also claim that they learn better when the teacher tells them what to do either suggests that there is a bias in the student responses that is either due to what Yackel and Cobb [26] call the "norms", which are also theorized as the "didactic contract" in Brousseau's terms [29] and which regulate the social interactions within the classroom, or because the role of the teacher as a leader is recognized by these students.

Table 4 summarizes the previous results in two categories (agree and disagree). The trend noted above can now be clearly seen.

**Table 4. Valid percent of students' answers participating in IGs and DLGs (Summary).**

| Item | Agree | Disagree |
|---|---|---|
| #1. We learn best when the teacher tells us what to do | 82,1 | 8,1 |
| #2. We learn more when we can express our own ideas | 74,4 | 7,2 |
| #3. Learning through discussion in class is confusing | 40,4 | 44 |
| #4. Sometimes learning in school is boring | 41,8 | 51,2 |
| #5. Learning in school is better when we have other adults to work with us | 76,9 | 13 |
| #6. I would rather think for myself than hear other people's ideas | 29,7 | 48,9 |
| #7. I enjoy learning when my friends help me | 78,6 | 11,4 |
| #8. It is good to hear other people's ideas | 78,3 | 12,7 |
| #9. Helping my friends has helped me to understand things better | 76,7 | 11,4 |
| #10. I am more confident about learning in school than I used to be | 77,2 | 8,9 |
| #11. I like discussing the books we read with the class | 67,2 | 20,9 |
| #12. At home, sometimes we talk about what I have been learning in school | 77,5 | 13,2 |

Items #4 and #10 are crucial to understanding the attitudes that students have towards learning. In the first case, half of the respondents contrarily claim that learning is a boring activity. If we assume that for a boy or a girl between 7 and 11 years old, defining an activity as fun or boring can be a clear way of indicating their attitude towards that activity, the fact that half of the students participating in the survey declare that learning is not boring suggests that their participation in doing mathematics in the IGs and DLGs makes these two activities in some way attractive to them.

On the other hand, another relevant aspect regarding the students' attitudes is the feeling of self-confidence. Previous research has provided much evidence that suggests the importance of this aspect in the attitudes that children can have towards learning [5]. Boys and girls who have confidence in themselves tend to show a clearly positive attitude towards learning. The data suggest that this is what happens when students participate within IGs or DLGs, i.e., three out of four children affirm that they feel more self-confident with regard to learning in school than they normally do. This result is relevant because it suggests that both IGs and DLGs have a clear impact on the positive transformation of attitudes towards learning. The data show that this is true for children in the three schools that participated in the survey, regardless of the country or the context in which they are located.

## Principal components analysis

The KMO test indicates whether the partial correlations between the variables are small enough to be able to perform a factorial analysis. Table 2 shows that in this case, the KMO test has a value of 0.517 for the students participating in IGs and 0.610 for the students participating in the DLGs, which allows us to assume (although with reservation) that we can perform a factor analysis to find the principal components explaining the variance. Bartlett's sphericity test (which contrasts the null hypothesis assuming that the correlation matrix is, in fact, an identity matrix, in which case we cannot assume that there are significant correlations between the variables) yields a critical value of 0.000 in both cases, which suggests that we can accept the null hypothesis of sphericity and, consequently, that we can think that the factorial model is adequate to explain de data.

After performing ANOVA several times, considering the several items in the tested models, we managed to find two models (one for the students who had participated in the IGs and another for those engaged in the DLGs) that explained more than half of the variance. Tables 5 and 6 introduce the obtained results.

As Tables 5 and 6 indicate, the items that are included in the SAM test contribute to better explaining the attitudes that the students participating in the study have regarding DLGs than those they have regarding the IGs. For the DLGs, we observed that there are four components above 1, explaining 74.227% of the variance. In contrast, in the case of IGs, we find only two

**Table 5. Total variance explained (students participating in the IGs).**

| Component | Initial Eigenvalues | | | Extraction Sums of Squared Loadings | | | Rotation Sums of Squared Loadings | | |
|---|---|---|---|---|---|---|---|---|---|
| | Total | % of Variance | Cumulative % | Total | % of Variance | Cumulative % | Total | % of Variance | Cumulative % |
| 1 | 1,465 | 36,631 | 36,631 | 1,465 | 36,631 | 36,631 | 1,439 | 35,970 | 35,970 |
| 2 | 1,063 | 26,570 | 63,201 | 1,063 | 26,570 | 63,201 | 1,089 | 27,230 | 63,201 |
| 3 | 0,868 | 21,692 | 84,893 | | | | | | |
| 4 | 0,604 | 15,107 | 100,00 | | | | | | |

Extraction method: Principal component analysis[a]

[a]The only cases used are those in which DLG or IG = IG in the analysis phase

**Table 6. Total variance explained (students participating in the DLGs).**

| Component | Initial Eigenvalues | | | Extraction Sums of Squared Loadings | | | Rotation Sums of Squared Loadings | | |
|---|---|---|---|---|---|---|---|---|---|
| | Total | % of Variance | Cumulative % | Total | % of Variance | Cumulative % | Total | % of Variance | Cumulative % |
| 1 | 1,803 | 22,540 | 22,540 | 1,803 | 22,540 | 22,540 | 1,446 | 18,076 | 18,076 |
| 2 | 1,148 | 14,352 | 36,891 | 1,148 | 14,352 | 36,891 | 1,418 | 17,730 | 35,806 |
| 3 | 1,059 | 13,239 | 50,131 | 1,059 | 13,239 | 50,131 | 1,132 | 14,156 | 49,962 |
| 4 | 1,045 | 13,063 | 63,193 | 1,045 | 13,063 | 63,193 | 1,059 | 13,232 | 63,193 |
| 5 | 0,883 | 11,034 | 74,227 | | | | | | |
| 6 | 0,768 | 9,598 | 83,826 | | | | | | |
| 7 | 0,696 | 8,699 | 92,525 | | | | | | |
| 8 | 0,598 | 7,475 | 100,00 | | | | | | |

Extraction method: Principal component analysis[a]

[a]The only cases used are those in which DLG or IG = DLG in the analysis phase

components above the value of 1, which together explain only 63.201% of the variance. This suggests that the SAM test is probably the best instrument to measure attitudes in the case of the DLGs.

The sedimentation graphs make it easier to visualize this result. In the left image of Fig 1 (the sedimentation graph obtained for the IGs), a clear inflection is observed from component 2. In contrast, in the case of the DLGs, the inflection occurs from component four and onward.

The matrix of components suggests that, in the case of the IGs, factor 1 is formed by the components that we can label as "active peer-support" (#9 "Helping my friends has helped me to understand things better") and "active listening" (#8 "It is good to hear other people's ideas"). Factor 2, on the other hand, is formed by the component of "participation" (#2 "We can learn more when we can express our own ideas"). For the IGs, the component "individualism" (#6 "I would rather think for myself than hear other people's ideas") is clearly the least explanatory (-0.616), which is a fact that seems to suggest that collaboration within the groups

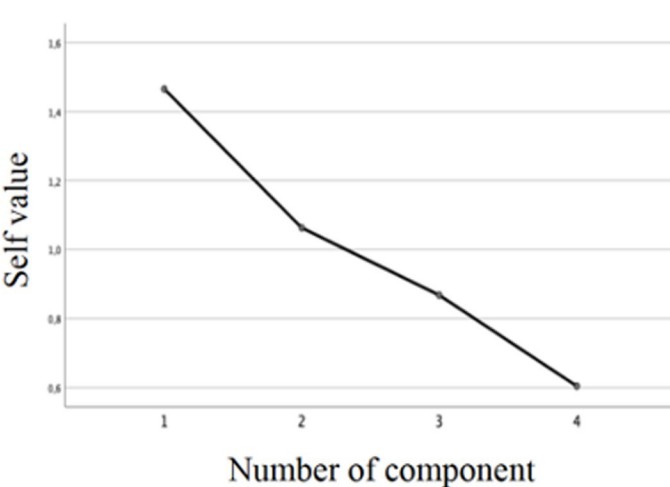 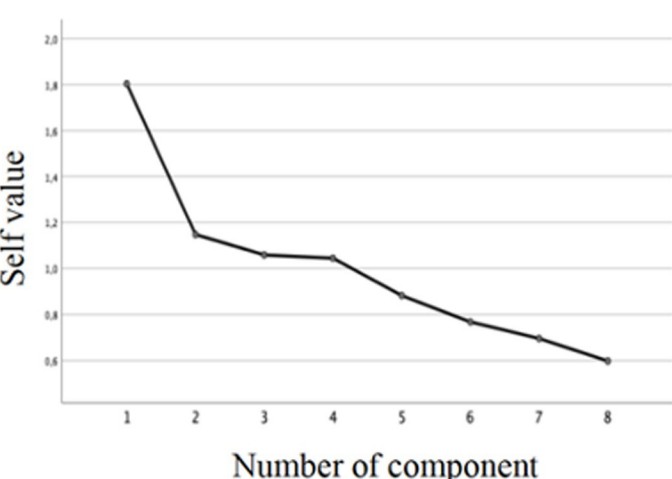

**Fig 1. Comparison between the sedimentation graphs corresponding to the analysis of principal components for IGs (image on the left side) and for DLGs (image on the right side).**

is a fundamental aspect of the learning dynamic occurring within them. Table 7 shows that the most explanatory factor of the variance is the first factor. On the other hand, factors 2, 3 and 4 are less important since their weights are almost irrelevant.

When we observe the results for the case of the DLGs, the component matrix (Table 8) indicates that factor 1 is formed mainly by the components of "positive discussion" (#11 "I like discussing the books we read with the class"), "self-confidence in school" (#10 "I am more confident about learning in school than I used to be), and "participation" (#2 "We can learn more when we can express our own ideas"). In contrast, Factor 2 contains a single component (#1 "We learn best when the teacher tells us what to do"). In the case of factor 3, the more explanatory component is the sixth component (#6 "I would rather think for myself than hear other people's ideas"). Finally, factor 4 includes the third component of the SAM test (#3 "Learning through discussing in class is confusing").

Fig 2 shows the graphs of the loading scores for each component in a rotated space, both for the IGs and the DLGs. The data confirm the interpretation of the previous tables. The graphs show that for the IGs (the left side of Fig 2), variables #8 and #9 tend to explain the maximum variance of factor 1, while in the case of DLGs, the three-dimensional component chart shows the two slightly differentiated groups of variables.

## Construction of subscales of attitudes towards learning in IGs and DLGs

The collected data allow us to think that the variables obtained with the SAM test may explain the attitudes that students have towards learning in the context of IGs and DLGs. However, according to previous theoretical models, it would seem plausible to assume that not all variables are equally precise in the explanation of the attitudes towards learning showed by the students interviewed in both contexts. For this reason, in this section, we compare two possible scales for each context (IGs and DLGs) to identify which variables would be more reliable in explaining those attitudes.

In the case of the IGs, we created two subscales. The first subscale (Tables 9–11) includes items #1 ("We learn best when the teacher tells us what to do"), #7 ("I enjoy learning when my friends help me") and #10 ("I am more confident about learning in school than I used to be"). In contrast, subscale 2 (Tables 12–14) incorporates items #2 ("We learn more when we can express our own ideas"), #5 ("Learning in school is better when we have other adults to work with us") and #11 ("I like discussing the books we read with the class"). Table 9 shows the Cronbach's alpha value for subscale 1, which is rather mediocre (0.412), while Table 12 indicates that subscale 2 is a much more reliable subscale (Cronbach's alpha of 0.794), suggesting that the subscale 2 works better than the first one to characterize the components explaining the results obtained within the IGs. The difference between the two subscales is that in the first one, the role of the teacher is not included, while in subscale 2, item #5 ("Learning in school is

**Table 7. Component matrix for IGs[a,b].**

|  | Component 1 | Component 2 |
|---|---|---|
| **#9. Helping my friends has helped me to understand things better** | 0,733 | -0,225 |
| **#2. We can learn more when we can express our own ideas** | 0,358 | 0,779 |
| **#8. It is good to hear other people's ideas** | 0,804 | 0,160 |
| **#6. I would rather think for myself than hear other people's ideas** | 0,392 | -0,616 |

Extraction method: Principal component analysis[a,b]

[a] 2 components extracted

[b] The only cases used are those for which DLG or IG = IG in the analysis phase

**Table 8. Component matrix for DLGs[a,b].**

|  | Component | | | |
|---|---|---|---|---|
|  | **1** | **2** | **3** | **4** |
| **#3. Learning through discussion in class is confusing** | -,068 | -,327 | ,467 | ,604 |
| **#11. I like discussing the books we read with the class** | ,667 | -,056 | ,070 | -,396 |
| **#1. We learn best when the teacher tells us what to do** | ,189 | ,782 | ,154 | -,130 |
| **#2. We can learn more when we can express our own ideas** | ,604 | -,416 | -,173 | ,107 |
| **#6. I would rather think for myself than hear other people's ideas** | ,053 | ,107 | ,842 | -,056 |
| **#8. It is good to hear other people's ideas** | ,519 | -,339 | ,238 | -,391 |
| **#10. I am more confident about learning in school than I used to be** | ,608 | ,295 | -,097 | ,347 |
| **#12. At home, sometimes we talk about what I have been learning in school** | ,558 | ,196 | -,088 | ,467 |

Extraction method: Principal component analysis[a,b]

[a] 4 components extracted

[b] The only cases used are those for which DLG or IG = DLG in the analysis phase

better when we have other adults to work with us") is the one that presents the highest correlation (0.613), as seen in Table 13, which shows the interitem correlation matrix for the variables of subscale 2.

Regarding DLGs, we also created two subscales, i.e., subscales 3 (Tables 15–17) and 4 (Tables 18–20). Subscale 3 is formed by variables #1 ("We learn best when the teacher tells us what to do"), #5 ("Learning in school is better when we have other adults to work with us"), #10 ("I am more confident about learning in school than I used to be") and #11 ('I like discussing the books we read with the class"). In contrast, subscale 4 includes variables #2 ("We learn more when we can express our own ideas"), #7 ("I enjoy learning when my friends help me"), #8 ("It is good to hear other people's ideas") and #9 ("Helping my friends has helped me to understand things better").

Table 15 shows the results for subscale 3 (DLGs), including a high Cronbach's alpha value (0.820), indicating that this subscale is a good proposal. According to data shown in Tables 16

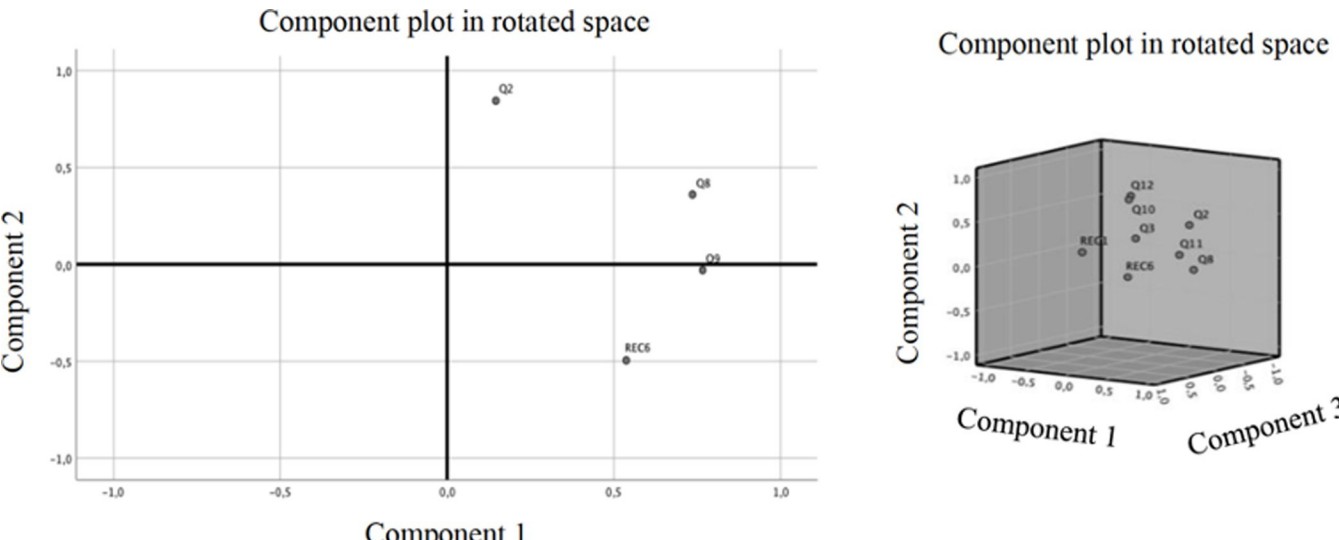

**Fig 2. Rotated component graphs for IGs (image on the left side) and for DLGs (image on the right side).**

**Table 9. Reliability statistics for subscale 1 (IGs).**

| Cronbach's Alpha | Cronbach's Alpha Based on Standardized Items | N of Items |
|---|---|---|
| 0,412 | 0,486 | 3 |

**Table 10. Inter-item correlation matrix for subscale 1 (IGs).**

| | #1 | #7 | #10 |
|---|---|---|---|
| **#1** | 1 | 0,316 | 0,31 |
| **#7** | 0,316 | 1 | 0,092 |
| **#10** | 0,31 | 0,092 | 1 |

**Table 11. Item-total statistics for subscale 1 (IGs).**

| | Scale Mean if Item Deleted | Scale Variance if Item Deleted | Corrected Item-Total Correlation | Squared Multiple Correlation | Cronbach's Alpha if Item Deleted |
|---|---|---|---|---|---|
| **#1** | 8,8261 | 2,332 | 0,381 | 0,18 | 0,095 |
| **#7** | 9,087 | 1,174 | 0,291 | 0,1 | 0,391 |
| **#10** | 8,3478 | 3,601 | 0,212 | 0,096 | 0,439 |

**Table 12. Reliability statistics for subscale 2 (IGs).**

| Cronbach's Alpha | Cronbach's Alpha Based on Standardized Items | N of Items |
|---|---|---|
| 0,794 | 0,799 | 3 |

**Table 13. Inter-item correlation matrix for subscale 2 (IGs).**

| | #2 | #5 | #11 |
|---|---|---|---|
| **#2** | 1 | 0,613 | 0,467 |
| **#5** | 0,613 | 1 | 0,628 |
| **#11** | 0,467 | 0,628 | 1 |

**Table 14. Item-total statistics for subscale 2 (IGs).**

| | Scale Mean if Item Deleted | Scale Variance if Item Deleted | Corrected Item-Total Correlation | Squared Multiple Correlation | Cronbach's Alpha if Item Deleted |
|---|---|---|---|---|---|
| **#2** | 8,7391 | 2,929 | 0,602 | 0,387 | 0,769 |
| **#5** | 8,913 | 2,992 | 0,722 | 0,526 | 0,626 |
| **#11** | 8,6087 | 3,613 | 0,604 | 0,406 | 0,757 |

**Table 15. Reliability statistics for subscale 3 (DLGs).**

| Cronbach's Alpha | Cronbach's Alpha Based on Standardized Items | N of Items |
|---|---|---|
| 0,820 | 0,830 | 4 |

**Table 16. Inter-item correlation matrix for subscale 3 (DLGs).**

|  | #1 | #5 | #10 | #11 |
|---|---|---|---|---|
| #1 | 1 | 0,537 | 0,259 | 0,608 |
| #5 | 0,537 | 1 | 0,582 | 0,781 |
| #10 | 0,259 | 0,582 | 1 | 0,528 |
| #11 | 0,608 | 0,781 | 0,528 | 1 |

and 17, the subscale 3 works better when component #10 is removed from the model (increasing Cronbach's alpha from 0,820 to 0,829). This result suggests that self-confidence is not a relevant component for attitudes towards participating in DLGs. Subscale 3 is better than subscale 4, which presents a Cronbach's alpha value of 0.767, which, although high, is lower than that found in subscale 3.

The clearest difference between both subscales (3 and 4) is that the first one (subscale 3) includes item #11 ("I like discussing the books we read with the class"), which is the one item more focused on the context of the DLGs. The other items are more related to the interaction among the students (Tables 18–20). However, the amount of correlation explained is lower than the amount explained by subscale 3. This fact may explain why subscale 3 is more reliable in measuring students' attitudes towards learning in social/interactional contexts, such as DLGs.

## Discussion and conclusions

Previous research in education has provided enough evidence to claim that attitudes have a relevant impact on learning [42–46]. Studies such as those of Fennema and Sherman [3] have confirmed almost half a century ago that the "affective factors (. . .) partially explain individual differences in the learning of mathematics" [3]. Currently, we know that the results of learning depend, to a certain extent, on the attitudes that students have towards it. When there is a clear resistance to school and school practices, it is more difficult for students to achieve good results. Aspects such as motivation or self-concept, for instance, are relevant to explaining a positive attitude towards learning. These aspects often appear to be correlated [19]. When a child has a poor self-concept as a student, s/he often feels very unmotivated to learn in school. The literature reports numerous cases of students who actively or passively resist or even refuse to make an effort to learn their lessons because they felt that they cannot learn anything. In contrast, when students' self-image is positive, then it is easier for them to learn. In those cases, the data provide evidence of positive correlations between learning achievements and attitudes. Rosenthal and Jacobson [5] called this type of behavior the Pygmalion Effect.

SEAs [7] such as IGs and DLGs are framed within the dialogic learning theory, one of whose main principals is that of transformation. As Freire [47] claimed, people "are beings of transformation, and not of adaptation." Education has the capacity to create opportunities for people to transform themselves. Drawing on the assumption that "education needs both

**Table 17. Item-total statistics for subscale 3 (DLGs).**

|  | Scale Mean if Item Deleted | Scale Variance if Item Deleted | Corrected Item-Total Correlation | Squared Multiple Correlation | Cronbach's Alpha if Item Deleted |
|---|---|---|---|---|---|
| #1 | 13,5800 | 6,160 | 0,545 | 0,392 | 0,816 |
| #5 | 13,1400 | 6,032 | 0,796 | 0,659 | 0,727 |
| #10 | 13,3200 | 6,143 | 0,518 | 0,365 | 0,829 |
| #11 | 13,4200 | 4,368 | 0,791 | 0,672 | 0,699 |

**Table 18. Reliability statistics for subscale 4 (DLGs).**

| Cronbach's Alpha | Cronbach's Alpha Based on Standardized Items | N of Items |
|---|---|---|
| 0,767 | 0,794 | 4 |

technical, scientific and professional training, as well as dreams and utopia" [47], SEAs integrate practices endorsed by the international scientific community to create real opportunities for learning for children. The data presented in the previous section suggest that children participating either in IGs or DLG have a clear positive attitude towards learning. Table 4 suggests that these children truly enjoy learning. Less than half of the participants say that learning at school is "boring" (41.8%). In contrast, almost eight out of ten children interviewed said they love learning (78.6%). The SAM test items, validated in previous studies [10,48,49], have been confirmed as components with which to measure children's attitudes towards learning. For the first time in the context of studying the impact of actions included within the SEAs [7], we have been able to identify (and measure) the positive relationship between implementing SEAs, i.e., boys and girls engaged in the IGs and/or the DLGs showed a clear positive attitude towards learning. We can therefore claim that in the context of SEAs, students show positive attitudes towards learning. The data analyzed suggest that participating within IGs and DLGs empower students to transform their own attitudes towards learning.

On the other hand, we know that social contexts have a powerful influence on people's attitudes. The context of positive empowerment, based on the idea of "maximum expectations" [50,51], is able to transform the attitudes that students have towards learning (especially those who are more resistant to learning and school). In contexts where school and school practices are not valued, children have to overcome the social tendency to openly show resistance against school (and everything that represents the school, such as teachers, attitudes of compliance with the school activities, norms, etc. ) and embrace a new tendency of valuing all these aspects. However, as previous studies framed within the symbolic interactionism approach have largely demonstrated [6,8], it is hard to turn against the social pressure of the group. We define our identity as a result of our interactions with others. If the group finds it attractive to resisting schooling, school norms and practices, then it is going to be difficult for individual students to achieve good academic results (unless they receive a different context from elsewhere) because they have to fight against the social pressure of not valuing school, in addition to the inherent difficulties of learning itself (in cognitive and curricular terms). In contrast, when the context is transformed (to adopt the terms of Freire and Flecha) and learning becomes a valued practice, children usually transform their attitudes, which previous research has correlated with successful learning achievement [14,15,33]. The data collected and discussed herein provide evidence for how changing the context (drawing on the two SEAs of IGs and DLGs) can transform students' attitudes towards learning. As we stated in the previous section, 78.6% of the students participating in the survey claimed that they like to learn after participating in either IGs or in DLGs. They claim that they like "when my friends help me." Along the same lines, 78.3% of the respondents said that "it is good to hear other people's

**Table 19. Inter-item correlation matrix for subscale 4 (DLGs).**

| | #2 | #7 | #8 | #9 |
|---|---|---|---|---|
| **#2** | 1 | 0,206 | 0,519 | 0,552 |
| **#7** | 0,206 | 1 | 0,516 | 0,474 |
| **#8** | 0,519 | 0,516 | 1 | 0,68 |
| **#9** | 0,552 | 0,474 | 0,680 | 1 |

**Table 20. Item-total statistics for subscale 4 (DLGs).**

| | Scale Mean if Item Deleted | Scale Variance if Item Deleted | Corrected Item-Total Correlation | Squared Multiple Correlation | Cronbach's Alpha if Item Deleted |
|---|---|---|---|---|---|
| **#2** | 13,6000 | 4,083 | 0,497 | 0,360 | 0,762 |
| **#7** | 13,9600 | 3,123 | 0,501 | 0,312 | 0,752 |
| **#8** | 13,9600 | 2,123 | 0,725 | 0,547 | 0,636 |
| **#9** | 13,800 | 3,417 | 0,721 | 0,543 | 0,661 |

ideas," while 76.7% claim that "helping my friends has helped me to understand things better." This type of answer clearly demonstrates that IGs and DLGs create a context in which learning is valued positively. Attitudes such as solidarity, willingness to help the other, friendship seem to indicate the preference for a context that is oriented towards learning rather than resisting it. Hence, transforming the context also changes how individuals recreate their own identities using different values as referents, which, drawing on Mead [6], is how identity creation works. The evidence collected herein suggests that IGs and DLGs work to increase students' academic performance because they transform the students' context; hence, students transform their own attitudes (as expected according the theory of symbolic interactionism).

By analyzing more in detail what happens in both the IGs and the DLGs, we have been able to verify that the attitudes that emerge among the students participating either in the IGs or the DLGs are slightly different. In the case of the IGs, the data collected reveal that children value much more the collaborative work with the rest of their classmates, as seen in the answers to items #8 ("It is good to hear other people's ideas") and #9 ("Helping my friends has helped me to understand things better"). These two items are the main components explained by the variance detected. On the other hand, in the case of the DLGs, the ability to express one's ideas is especially valued. In this case, the variance is explained above all by items #2 ("We learn more when we can express our own ideas"), #10 ("I am more confident about learning in school than I used to be") and #11 ("I like discussing the books we read with the class"). The last component (#11) clearly belongs to a context similar to that of the DLGs. However, the two previous components (#2 and #10) suggest that participation in DLGs is related to the development of a positive self-image as learner. The chi-square test indicates that the correlations are significant in both cases. Therefore, the data obtained suggest that participating in IGs and/or DLGs is related to showing positive attitudes towards learning (both as an individual and as member of the group, i.e., in a social sense).

On the other hand, when analyzing the reliability of the results, it can be verified that in the case of the IGs, the most important correlation appears in the case of item #5 ("Learning in school is better when we have other adults to work with us"). This finding is very relevant since it constitutes empirical evidence of something that Vygotsky already suggested when he proposed his concept of ZPD, which is that in order for the process to work, there must be an adult or a more capable peer to help those who are learning achieve what they can with the support of these adults who act as facilitators. The difference between IGs and other collaborative learning groups is exactly that, i.e., in the IGs, there is always an adult who dynamizes the activity (who does not provide the answers but encourages the children to engage in a dialogic interaction [15]).

Regarding the DLGs, the most important correlation appears in the case of item #11 ("I like discussing the books we read with the class"), which is an aspect that makes sense in the context of the gatherings. Children affirm that they like to read books together with their other classmates. As we know, this activity has clear advantages from the point of view of the development of reading understanding [41,52].

A surprising finding is the high response rate to item #1 ("We learn best when the teacher tells us what to do"). This would seem to be inconsistent with using IGs or DLGs, where the role of the teacher is rather marginal or passive (the teacher organizes the activity but does not give answers, and they explain the academic content such as in a master's class, etc.). Perhaps a possible reason to explain this result is that the school, as an institution, is characterized by a series of social norms [26]. Waiting for teachers' directions is part of those norms. It is assumed that when attending the school, we must pay attention to what the teacher says. This idea corresponds to the social image of the teacher as a transmitter of knowledge, which is part of the social norm characterizing the school institution. It is possible that even though the children participating in this study have engaged in IGs and DLGs, there are not excluded from the norms of the social context, so that their attitudes are tinged with them.

We can therefore conclude that the SAM test demonstrates that children who participate in IGs and/or DLGs clearly show positive attitudes towards learning after participating in these two SEAs. Perhaps this is one of the fundamental variables explaining the successful learning results that other studies have found among children using SEAs [11–15].

## Future implications

This research confirms some aspects of learning, while it leaves others open for further study. We have observed that children who participate in SEAs show positive attitudes towards learning. However, what we do not know (yet) is whether it is the use of these SEAs that explains why these children show these attitudes or if the transformation lies in other reasons. To clarify this lack of information, it is necessary to conduct further experimental research comparing groups of students using SEAs and other groups of students using other types of educational actions.

On the other hand, the data that we have discussed herein suggest that there is a social component that has a critical influence on the type of attitudes that students report in the survey. According to the criteria of how the IGs and DLGs work, solidarity, interaction, and sharing seem to explain why these children develop positive learning attitudes. However, it would be interesting to continue with this line of research to see if this outcome also presents when other educational actions are used in which the principals of action are different (when they are centered on the individual, for example).

Finally, evidence seems to support the statement that the successful academic performance of children who participate in IGs and DLGs is explained by the fact that participating in these two types of SEAs transforms the children's context to a positive orientation towards learning. Indeed, the results are hopeful. However, we need to further replicate this study to confirm (or refute) that statement. In any case, confirming that statement and covering the preceding research questions presents the clear implication that teachers have to put their effort into designing their lessons, as how they organize their classes truly encourages students' learning.

## Supporting information

**S1 File. SAM questionnaire: What I think about learning in school.**
(PDF)

**S2 File. Dataset.**
(SAV)

## Author Contributions

**Conceptualization:** Rocío García-Carrión, Linda Hargreaves.

**Data curation:** Javier Díez-Palomar.

**Formal analysis:** Javier Díez-Palomar, Linda Hargreaves.

**Investigation:** Rocío García-Carrión, Linda Hargreaves, María Vieites.

**Methodology:** Javier Díez-Palomar, Rocío García-Carrión, Linda Hargreaves.

**Project administration:** María Vieites.

**Supervision:** Linda Hargreaves.

**Validation:** Javier Díez-Palomar, Linda Hargreaves.

**Writing – original draft:** Javier Díez-Palomar.

**Writing – review & editing:** Javier Díez-Palomar, Rocío García-Carrión, Linda Hargreaves, María Vieites.

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
