## [Decision Letter · Decision Letter 0]

30 Jul 2020

PONE-D-20-08828

Transforming Students’ Attitudes Towards Learning Through the Use of Successful Educational Actions

PLOS ONE

Dear Dr. Diez-Palomar,

Thank you for submitting your manuscript to PLOS ONE. After careful consideration, we feel that it has merit but does not fully meet PLOS ONE’s publication criteria as it currently stands. Therefore, we invite you to submit a revised version of the manuscript.

That revision is truly a minor one: according to the PLOS ONE data reporting policy, authors are required to indicate where the study data may be found. Therefore, please add to your indication that “the data used in this article comes (sic) from the research project *SEAs4All – Schools as Learning Communities in Europe*” the URLs, accession numbers or DOIs if your data are held or will be held in a public repository. If that does not apply but you are able to provide details of access elsewhere, with or without limitations, please do so. For details, please see the journal’s data reporting guidelines at https://journals.plos.org/plosone/s/submission-guidelines#loc-data-reporting.

A rebuttal letter. You should upload this letter as a separate file labeled 'Response to Reviewers'.A marked-up copy of your manuscript that highlights changes made to the original version. You should upload this as a separate file labeled 'Revised Manuscript with Track Changes'.An unmarked version of your revised paper without tracked changes. You should upload this as a separate file labeled 'Manuscript'.

We look forward to receiving your revised manuscript.

Kind regards,

Christian Stamov Roßnagel

Academic Editor

PLOS ONE

2.In your Data Availability statement, you have not specified where the minimal data set underlying the results described in your manuscript can be found. PLOS defines a study's minimal data set as the underlying data used to reach the conclusions drawn in the manuscript and any additional data required to replicate the reported study findings in their entirety. All PLOS journals require that the minimal data set be made fully available. For more information about our data policy, please see http://journals.plos.org/plosone/s/data-availability.

3. We note you have included a table to which you do not refer in the text of your manuscript. Please ensure that you refer to Table 1, 8, 10, 11, 14, 16, 17, 18, 19 and 20 in your text; if accepted, production will need this reference to link the reader to the Table.

4. Your ethics statement must appear in the Methods section of your manuscript. If your ethics statement is written in any section besides the Methods, please move it to the Methods section and delete it from any other section. Please also ensure that your ethics statement is included in your manuscript, as the ethics section of your online submission will not be published alongside your manuscript.

Reviewers' comments:

Reviewer's Responses to Questions

**Comments to the Author**

1. Is the manuscript technically sound, and do the data support the conclusions?

Reviewer #1: Partly

Reviewer #2: Yes

Reviewer #3: Yes

2. Has the statistical analysis been performed appropriately and rigorously? 

Reviewer #1: I Don't Know

Reviewer #2: Yes

Reviewer #3: Yes

3. Have the authors made all data underlying the findings in their manuscript fully available?

Reviewer #1: Yes

Reviewer #2: Yes

Reviewer #3: Yes

4. Is the manuscript presented in an intelligible fashion and written in standard English?

Reviewer #1: Yes

Reviewer #2: Yes

Reviewer #3: Yes

5. Review Comments to the Author

Reviewer #1: (No Response)

Reviewer #2: The paper has a sound theoretical introduction later discussed and enriched with empirical quantitative data. It is remarkable that the authors clearly explain what was already known about the two educational actions explored and which is the specific contribution of this article. Data is collected in three different schools (in UK and Italy), whose contexts are sufficiently explained. The detailed description of the methodology and results is one of the strengths of the paper.

The authors confirm that all data underlying the findings described in their manuscript are fully available without restriction, but they should specify where the data can be found.

Reviewer #3: This research supposes an advance for the educative field, it also enables the improvement of concrete practical education. In order to know about the impact of interactive groups (IG) and dialogic literary gatherings (DLG) in relation to attitudes and academic achievement, this research is original and unpublished.

The research also shows to be well documented, both theoretically and with the latest research carried out. The article has consistency in all sections, which provides coherence and reliability of the study.

6. PLOS authors have the option to publish the peer review history of their article (what does this mean?). If published, this will include your full peer review and any attached files.

Reviewer #1: No

Reviewer #2: No

Reviewer #3: No

---

## [Author Response · Author response to Decision Letter 0]

16 Sep 2020

Dear Editor, 

Here we attach the article “Transforming Students’ Attitudes Towards Learning Through the Use of Successful Educational Actions” for your consideration, responding to requests that you sent us by July 30th, 2020. 

In your message, you requested us to include / check / review the following aspects: 

(1) “Please ensure that your manuscript meets PLOS ONE's style requirements, including those for file naming. The PLOS ONE style templates can be found at: 

https://journals.plos.org/plosone/s/file?id=wjVg/PLOSOne_formatting_sample_main_body.pdf and https://journals.plos.org/plosone/s/file?id=ba62/PLOSOne_formatting_sample_title_authors_affiliations.pdf”

We have edited the whole article, according to the PLOS ONE style requirements. 

(2) “In your Data Availability statement, you have not specified where the minimal data set underlying the results described in your manuscript can be found. PLOS defines a study's minimal data set as the underlying data used to reach the conclusions drawn in the manuscript and any additional data required to replicate the reported study findings in their entirety. All PLOS journals require that the minimal data set be made fully available. For more information about our data policy, please see http://journals.plos.org/plosone/s/data-availability. 

We will update your Data Availability statement to reflect the information you provide in your cover letter.”

We added the following sentence in the methodological section: “The dataset has been submitted to this journal as supporting data for public use.” In addition, we are attaching a file named S2 File 2 containing the dataset in SPSS format. 

(3) “We note you have included a table to which you do not refer in the text of your manuscript. Please ensure that you refer to Table 1, 8, 10, 11, 14, 16, 17, 18, 19 and 20 in your text; if accepted, production will need this reference to link the reader to the Table.” 

All tables have been reviewed and referred along the main text of the article. Additional / further explanations have been added when needed: 

• (line 397-398) Table 7 shows that the most explanatory factor of the variance is the first factor. On the other hand, factors 2, 3 and 4 are less important since their weights are almost irrelevant.

• (line 437-438) suggesting that the subscale 2 works better than the first one to characterize the components explaining the results obtained within the IGs.

• (line 459-461) According to data show in Tables 16 and 17, the subscale 3 works better when component #10 is removed from the model (increasing Cronbach’s alpha from 0,820 to 0,829). This result suggest that self-confidence is not a relevant component for attitudes towards participating in DLGs.

(4) “Your ethics statement must appear in the Methods section of your manuscript. If your ethics statement is written in any section besides the Methods, please move it to the Methods section and delete it from any other section. Please also ensure that your ethics statement is included in your manuscript, as the ethics section of your online submission will not be published alongside your manuscript.”

The ethics statement has been placed in a section besides the methodology, right after that. 

We would like to thank you very much all the comments that have improved significantly the quality of this article.

Sincerely, 

Javier Díez-Palomar, Rocío García-Carrión, Linda Hargreaves and María Vieites

---

## [Editor Report · Decision Letter 1]

24 Sep 2020

Transforming Students’ Attitudes Towards Learning Through the Use of Successful Educational Actions

PONE-D-20-08828R1

Dear Dr. Diez-Palomar,

Thank you for your careful revision of your manuscript.

We’re pleased to inform you that your manuscript has been judged scientifically suitable for publication and will be formally accepted for publication once it meets all outstanding technical requirements.

Kind regards,

Christian Stamov Roßnagel

Academic Editor

PLOS ONE
---

## [Editor Report · Acceptance letter]

28 Sep 2020

PONE-D-20-08828R1 

Transforming Students’ Attitudes Towards Learning Through the Use of Successful Educational Actions 

Dear Dr. Díez-Palomar:

I'm pleased to inform you that your manuscript has been deemed suitable for publication in PLOS ONE. Congratulations! Your manuscript is now with our production department. 

Kind regards, 

on behalf of

Mr Christian Stamov Roßnagel 

Academic Editor

PLOS ONE